# Management of preterm birth using protocols in a low resource setting

**Jackline Akello**[1,2]*, **Fatuma Namusoke**[1,2], **Godfrey Alia**[2], **Savio Mwaka**[3]*

**1** Department of Obstetrics and Gynecology, College of Health Sciences, Makerere University, Kampala, Uganda, **2** Mulago Specialized Women's and Neonatal National Referral Hospital, Kampala, Uganda, **3** Health Systems Strengthening Cluster, Walimu, Kampala, Uganda

* akellojack@gmail.com (JA); saviomwaka@gmail.com (SM)

## Abstract

### Background

Preterm birth is the leading cause of neonatal deaths and the second leading cause of death in children under five after pneumonia. The study aimed at improving the management of preterm birth through the development of protocols for standardization of care.

### Methods

The study was conducted in Mulago National Referral Labor ward in two phases. A total of 360 case files were reviewed and mothers whose files had missing data interviewed for clarity for both the baseline audit and the re-audit. Chi squares were used to compare results for the baseline and the re-audit.

### Results

There was significant improvement in four parameters out of the six that were used to assess quality of care and these were 32% increase in administration of Dexamethasone for fetal lung maturity, 27% increase in administration of Magnesium Sulphate for fetal neuroprotection and 23% increase in anti-biotic administration. A 14% reduction noted in patients who received no intervention. However, there was no change in the administration of Tocolytic.

### Conclusion

The results of this study have shown that protocols standardize care and improve the quality of care in preterm delivery to optimize outcomes.

## Introduction

Preterm birth is birth of the baby before 37 completed weeks of gestational age [1]. Preterm birth is prevalent in many countries with an upward trend currently accounting for about 11%

**Data Availability Statement:** http://makir.mak.ac.ug, Criterion based audit on the management of preterm birth in Mulago National Referral hospital.

**Funding:** The author(s) received no specific funding for this work.

**Competing interests:** The authors have declared
that no competing interests exist.

of deaths of children under the age of five years [2]. In Sub Saharan Africa and South Asia, 60% of perinatal deaths are due to prematurity and in Uganda, 31% of its neonatal deaths are due to prematurity [3–5].

As a result of the increasing contribution of neonatal deaths to overall child mortality, it is critical to address the determinants of poor outcomes related to preterm birth through interventions delivered to the mother before or during pregnancy, and to the preterm infant after birth. However, the most beneficial set of maternal interventions are those that are aimed at improving outcomes for preterm infants when preterm birth is inevitable which include tocolysis, antenatal corticosteroids, magnesium sulfate and antibiotic prophylaxis [6–8].

These interventions form the basis for setting standard guidelines or protocols to guide and improve management of different obstetric complications with the aim of reducing mortality and morbidity due to improved care. Indeed, the need for protocols to standardize care is highlighted by studies in the same setting, by Alia et al on management of severe preeclampsia [9] and kayiga et al on the management of obstructed labor [10].

The study therefore aimed to improve management of preterm birth in Mulago National Referral hospital through development of a protocol to standardize care through an audit process.

## Methods

This was a facility-based study in Mulago National Referral hospital labor ward from September 2018 to February 2019. Mulago National Referral Hospital had its department of Obstetrics and Gynecology housed in Kawempe division due to the ongoing renovation of the main hospital. The department has maternal fetal unit, labor, postnatal gynecological and oncological wards as well as outpatient clinics. It also has special care unit for managing neonates with birth complications and prematurity.

On average, there are about 60–80 deliveries a day in the labor ward and about 3–5 of these are due to preterm labor. The labor suite has an in-charge who oversees the daily running of the ward with day and night shifts of at least one obstetrician, 5 senior house officers, 3 junior house officers and five midwives who manage the ward.

Mothers with anticipated preterm delivery are admitted in the high-risk labor ward and babies delivered preterm are resuscitated and immediately taken to special care unit for management by the pediatric group. The mothers who have had preterm birth are taken to postnatal ward where they are either discharged home after twenty-four hours or sent to mother's club which is a special room for post-natal mothers with babies in SCU.

It was consecutive sampling where all files and mothers who met the inclusion criteria were assessed and their records reviewed with the checklist until the sample size was attained.

On the basis of the official figures from Mulago for FY2016/17, Mulago Hospital receives on average five pre-terms daily making an estimate of 150 pre-terms a month. With the study split in two phases of two months each, the sample size formula adjusted for small populations by Cochran was used

$$n* = ((z_{\frac{\alpha}{2}})^\wedge 2\, PQ)/D^2$$

$$n > (n*/(1 + (n* - 1)/N))$$

This gives a minimum sample size of 169 patients in each phase
Only mothers who had preterm birth and had either

1. Ultra sound scan.

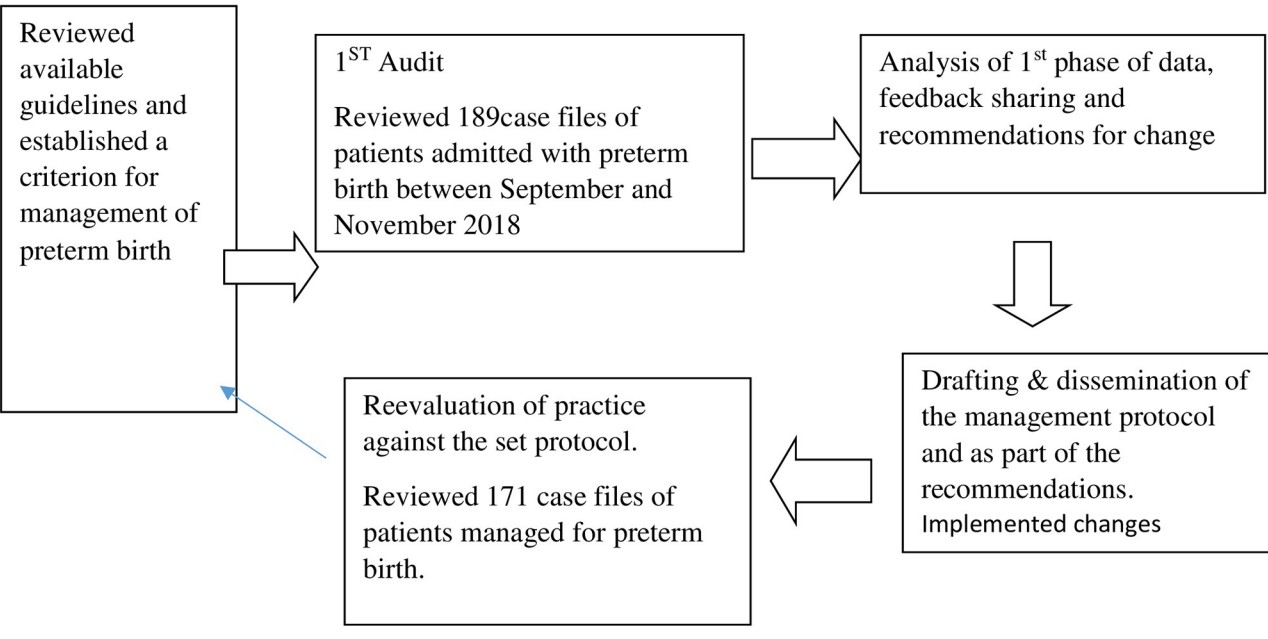

**Fig 1. Audit cycle.**

2. known LNMP, or

3. Alive baby for ballards score to ascertain gestational age, were included in the study.

A pilot study was done for a period of two weeks to assess feasibility of data collection, the tool and analysis plan.

It was prospective observational study done in a step wise manner as shown in Fig 1.

The Delphi technique was applied by a team of experts from both Makerere University and Mulago National Referral hospital maternal fetal specialists who derived the components of preterm birth care as:

1. Antenatal corticosteroids administration if gestational age < 34 weeks. This included 4 doses of dexamethasone for GA < = 34 weeks and a single dose of Betamethasone for GA between 35–36 completed weeks.

2. Magnesium sulphate for fetal neuroprotection for GA <34 weeks

3. Antibiotic administration in cases of PPROM

4. Tocolysis if cervical dilation <4cm to enable corticosteroid administration

The mothers who met the inclusion criteria were then recruited for the study and a total of 189 case files were reviewed and compared against the set standards for a period of two months. Participants were also interviewed for details where there was need for clarity of documentation of clinical findings.

The results of the 1st audit were analyzed and discussed in special staff meeting where all cadres involved in preterm birth management were adequately represented. These included the hospital clinical head, the consultants and or specialists, senior house officers (residents), intern doctors and nurses and midwives that run the maternity department. The biggest gap in care identified was the lack of national or institutional guideline or protocol to standardize care for preterm birth.

Two months later, the audit was repeated following the dissemination of the protocol in another staff meeting that was also adequately attended by the same team who participated in the first feedback meeting.

Data was entered through Kobo collect, an online data collection application, then exported and cleaned using STATA 14 by the principal investigator and the statistician. Access to the data was limited using password on Kobo collect and the researcher's computer. The chi square test was used as the unit of analysis to compare both audits and statistical significance was established with a p value of less than 0.05% (CI of 95%).

## Results

A total of 360 files were reviewed. In the first audit, 189 were reviewed. The second audit comprised of 171 files.

Table 1 is a presentation of the baseline characteristics of the patients.

### Results from baseline audit

Since Mulago National Referral hospital had no institutional guidelines or criteria to manage anticipated preterm birth and also given the fact that there was no national guideline for management of anticipated preterm delivery, the initial criteria followed for the baseline audit was set up using the Delphi technique, where a team of experts did a desk review of several international guidelines and came up with a recommendation on the set of interventions for anticipated preterm delivery.

Table 2 presents the findings from the baseline audit.

A feedback session was held with the team that discussed the initial audit inclusive of hospital administration, specialists, Senior House Officers, interns and midwives who all participate in the management of these patients. The team identified not being knowledgeable

**Table 1. Baseline characteristics of 360 patients with preterm birth managed for both phases of the audit in Mulago National Referral Hospital labor ward.**

|  | First Audit (n = 189) | 95%CI | Second Audit (n = 171) | 95%CI | P |
|---|---|---|---|---|---|
| **Gestation** |  |  |  |  |  |
| LNMP | 51.9% | (44.9, 59.1) | 74.3% | (67.4, 80.6) | **0.000** |
| US Scan | 42.3% | (35.0, 49.0) | 17.5% | (12.2, 23.8) | **0.000** |
| Ballad Score | 5.8% | (2.6, 9.4) | 8.2% | (3.9, 12.1) | 0.456 |
| **Mode of Delivery** |  |  |  |  |  |
| C-Section | 43.4% | (35.9, 50.1) | 29.8% | (23.1, 36.9) | 0.011 |
| Vaginal Delivery | 56.6% | (49.9, 64.1) | 70.2% | (63.1, 76.9) | 0.011 |
| **Outcome** |  |  |  |  |  |
| Alive | 85.2% | (79.9, 90.1) | 85.4% | (79.6, 90.4) | 1.000 |
| Early Neonatal Death | 5.8% | (2.6, 9.4) | 11.1% | (6.3, 15.7) | 0.087 |
| Fresh Still Birth | 5.8% | (2.6, 9.4) | 2.9% | (0.4, 5.6) | 0.174 |
| Macerated Stillbirth | 3.2% | (0.6, 5.4) | 0.6% | (0.5, 2.5) | 0.181 |
| **Pre-term categorization** |  |  |  |  |  |
| Febrile Illness | 27.5% | (21.6, 34.4) | 2.9% | (0.4, 5.6) | 0.894 |
| PPROM | 55.0% | (47.9, 62.1) | 54.4% | (46.5, 61.5) | **0.000** |
| Pre(Eclampsia | 34.9% | (28.2, 41.8) | 21.1% | (14.9, 27.1) | **0.003** |
| Cervical Dilation <4cm | 37.0% | (30.1, 43.9) | 35.7% | (28.8, 43.2) | 0.844 |
| **Weeks of Gestation** |  |  |  |  |  |
| < = 34 Weeks | 74.1% | (67.7, 80.3) | 76.6% | (70.7, 83.3) | 0.509 |

**Table 2. Findings from baseline audit as per guidelines by Delphi technique for preterm birth management.**

| Audit parameter | Intervention offered | Total that required the intervention | Adherence |
|---|---|---|---|
| Cervical Dilation < = 4cm received Dexamethasone MgSO4 and Tocolytic | 20 | 70 | 29% |
| On ward >48 hours and received 4 doses of dexamethasone | 56 | 71 | 79% |
| < = 34 weeks received dexamethasone | 69 | 140 | 49% |
| < = 32weeks received MgSO4 | 28 | 100 | 28% |
| Non-Pre(eclampsia) received MgSO4 | 0 | 123 | 0% |
| PPROM receiving antibiotics | 48 | 104 | 46% |
| No intervention (Dexa, MgSO4, Tocolytic nor Antibiotics) | 55 | 189 | 29% |
| > = 35 receiving dexamethasone | 17 | 49 | 35% |
| < = 34 with PPROM and febrile illness receiving dexa, Tocolytic and antibiotics | 3 | 34 | 9% |

about general preterm care as a major hindrance to achieving the desired quality of care. This was particularly faced by the interns and some senior house officers as well as the midwives who rely on the doctor's prescription to administer treatment. This is compounded by the lack of a clear institutional guideline or protocol on how to manage preterm birth.

The team therefore agreed on development of an institutional protocol and a re -audit based on the protocol. Standard of care meant achieving 100% in all parameters of the audit.

Table 3 presents the findings from the baseline and second audit as per the developed protocol.

There was a 32% increase 95% CI (42%,21%) in the patients who received Dexamethasone for fetal lung maturity significant at p<0.001, a 27% increase 95% CI (38%,16%) in the patients that received Magnesium Sulphate for fetal neuroprotection significant at p<0.001, a 23% increase in the patients with PPROM that received anti-biotic 95% CI (36%,9%) significant at p<0.05 and a 14% reduction in those who received no intervention at 95% CI (5%, 24%) which was significant at p>0.05. However, there was no change in the administration of Tocolytic.

## Discussion

Protocols / guidelines improve the quality of medical care, including the procedures used for diagnosis and treatment, the use of resources and the resulting outcome and quality of life for the patient [11]. When used consistently, clinical care protocols iron out variations in practice and can be easily used to track performance. The study aimed at assessing the extent to which a protocol can improve quality of care given to women with preterm birth through an audit process. The results show that the implementation of a protocol in the management of preterm birth led to measurable improvements in 4 audit parameters out of the six (administration of dexamethasone, magnesium sulphate, antibiotic, and those who received no intervention).

**Table 3. A comparison of the baseline and second audit for preterm birth management as per the developed protocol.**

| Audit Parameters | First Audit | Second Audit | Risk Diff | 95% CI | P |
|---|---|---|---|---|---|
| < = 34 Weeks that received Dexamethasone | 49% | 81% | +32% | (42,21) | **0.000** |
| < = 34 Weeks that received Mgso4 | 28% | 55% | +27% | (38,16) | **0.000** |
| PPROM that received anti-biotic | 46% | 69% | +23% | (36,9) | **0.001** |
| < = 34 Weeks with Cervical Dilation< = 4cm that received a Tocolytic | 3% | 4% | +1% | (6, -3) | 0.549 |
| Received no intervention | 35% | 20% | -14% | (5,24) | **0.002** |
| On ward >48 days and received 4 doses of dexamethasone | 79% | 89% | +10% | (-23,4) | 0.179 |

The 32% increment in the administration of Dexamethasone for fetal lung maturity for Gestational could have been achieved due to the fact that this is already in the Uganda National Clinical Guidelines and most clinicians just need constant reminders in the form of Job aids to ease prescription [12]. However, despite the 27% improvement in the administration of magnesium sulphate for fetal neuroprotection, most health workers are still un comfortable prescribing it despite the available literature probably due to the fact that it is not yet included in the national guidelines for that purpose. The administration of tocolytic stagnated at 4% due to majorly fear of combined effect of Magnesium sulphate which is also a weak tocolytic with another tocolytic agent. It is also worth noting that despite proper prescription, some patients still did not receive the rightful treatment as seen in the 20% who received no intervention and the 11% who were on ward for more than 48 hours and still did not receive all the four doses of dexamethasone. Significant to note is also the increase in the early neonatal deaths in the second audit which points to institutional challenges inclusive of shortage of surfactant, antibiotics and oxygen for managing preterm in the special neonatal care units which are periodic depending on the availability of supplies and were out of stock at that time.

Though there was generally overall improvement in the care given to mothers with anticipated preterm birth, the optimal quality of care was not met. Several factors led to the failure to achieve the recommended standard of care and these included lacks of team work as well as institutional shortages.

In a similar study by Kayiga et al, about the management of obstructed labor, they noted significant improvement in two out of the four audit parameters and similar factors hindering achievement of optimal standard of care were identified [10].

Another similar study by Ononge et al, in the management of preeclampsia, in the same setting had similar results with significant improvement noted after guidelines were put in place as well as purchase of additional supplies to curb institutional shortages. The initial audit of this study also noted that standards were rarely achieved [14].

The use of clinical care protocols to improve obstetric care is not only feasible and plausible for developing countries with limited resources, but also backed up by researches in many settings with similar findings. A study done in Ibadan, Nigeria by Hunyinbo KI et al on Evaluation of Criteria-Based Clinical Audit in Improving Quality of Obstetric Care in a Developing Country Hospital, generally high lights the need for institutional protocols and guidelines to improve care as well as addressing the basic resources for health care [13–17].

Clinical care protocols serve as a reference point for clinical decision making and performance improvement and are central to the practice of evidence-based medicine as well as transforming solid evidence into meaningful patient care [18].

Indeed, the standardization of practice to improve quality of care is important in optimizing patient outcomes. Protocols should therefore be institutionalized as guides for the care of patients based on most current literature and global standards [19].

## Conclusion

The results of this study have shown that protocols standardize care and improve quality of care in preterm birth to optimize outcomes.

## Acknowledgments

Special thanks to the administration of Makerere University, college of Health Sciences, directorate of Obstetrics and Gynecology for guidance, support and supervision.

We would like to extend our sincere gratitude to all Mulago hospital staff and patients without whom this research would not be possible and finally to our editor, Dr Quraish Sserwanja for all the copyedits done.

## Author Contributions

**Conceptualization:** Jackline Akello, Fatuma Namusoke, Godfrey Alia, Savio Mwaka.

**Data curation:** Jackline Akello.

**Formal analysis:** Jackline Akello, Savio Mwaka.

**Investigation:** Jackline Akello, Fatuma Namusoke, Savio Mwaka.

**Methodology:** Jackline Akello, Fatuma Namusoke, Savio Mwaka.

**Project administration:** Jackline Akello, Fatuma Namusoke.

**Resources:** Jackline Akello, Godfrey Alia.

**Software:** Savio Mwaka.

**Supervision:** Jackline Akello, Fatuma Namusoke, Godfrey Alia, Savio Mwaka.

**Validation:** Jackline Akello, Fatuma Namusoke.

**Writing – original draft:** Jackline Akello.

**Writing – review & editing:** Jackline Akello, Savio Mwaka.

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
