## [Decision Letter · Decision Letter 0]

15 Feb 2023

PONE-D-23-00862Management of preterm birth using protocols in a low resource setting.PLOS ONE

Dear Dr. Jackline Akello

Thank you for submitting your manuscript to PLOS ONE. After careful consideration, we feel that it has merit but does not fully meet PLOS ONE’s publication criteria as it currently stands. Therefore, we invite you to submit a revised version of the manuscript that addresses the points raised during the review process.

We look forward to receiving your revised manuscript.

Kind regards,

Lumaan Sheikh, FRCOG

Academic Editor

PLOS ONE

Journal Requirements:

- https://www.ncbi.nlm.nih.gov/books/NBK321160/pdf/Bookshelf_NBK321160.pdf?

https://journals.lww.com/greenjournal/Fulltext/2019/10000/Clinical_Guidelines_and_Standardization_of.50.aspx?

In your revision ensure you cite all your sources (including your own works), and quote or rephrase any duplicated text outside the methods section. Further consideration is dependent on these concerns being addressed.

Reviewers' comments:

Reviewer's Responses to Questions

**Comments to the Author**

1. Is the manuscript technically sound, and do the data support the conclusions?

Reviewer #1: Yes

Reviewer #2: Yes

2. Has the statistical analysis been performed appropriately and rigorously? 

Reviewer #1: Yes

Reviewer #2: Yes

3. Have the authors made all data underlying the findings in their manuscript fully available?

Reviewer #1: Yes

Reviewer #2: Yes

4. Is the manuscript presented in an intelligible fashion and written in standard English?

Reviewer #1: No

Reviewer #2: Yes

5. Review Comments to the Author

Reviewer #1: The authors have chosen a very important topic which is specially pertinent to LMIC

The objective of the study is well defined

The methodology is overall well written but the numbers mentioned as numerator and denominator in Table no. 2 are not clear. This should be reworded for clarity.

It is not clear what is meant by the following sentence on page no. 8," Since there was no documented protocol on management of pre-term birth at Mulago Hospital labor ward, the baseline audit was conducted as per the components derived by the team of experts". This needs to reworded for clarity.

Although second audit showed improvement in quality of care in patients with preterm labour/delivery, the authors have also discussed in length the reasons for inability to meet all the parameters to the desired extent. It would be informative if authors could give suggestions on how to improve the deficiency.

There are several grammatical errors that need to be corrected.

Reviewer #2: The rationale in abstract and introduction should be similar. Certain grammatical mistakes are there. Page 4 4th paragraph instead of using word "basing" write "on the basis of" or use another word. Ideally referring to table write the number in numerical form, Table 2 rather than table two.

In the results the outcomes table, there is early neonatal death in 1st audit that is given 5.8% the second audit done after observing protocol ,the early neonatal death increased to 11% which is quite significant. What were the reasons please mention in discussion.

In methods you have mentioned in page 6 that participants were interviewed, was consent taken as its not mentioned how was the consent taken please mention. You have stated that staff were given information of the first audit outcome and in the second audit you mentioned the findings were discussed in another meeting .Mention who were the staff that received the information of both audit results? were they the same as in the first audit or a different group.Page 9 first paragraph rephrase the paragraph, long sentence used as well as structure of sentence is unclear.

Conclusion should reflect your study findings and not a general opinion or suggestion.

6. PLOS authors have the option to publish the peer review history of their article (what does this mean?). If published, this will include your full peer review and any attached files.

Reviewer #1: **Yes: **Azra Amerjee

Reviewer #2: **Yes: **Ambareen Khan

---

## [Author Response · Author response to Decision Letter 0]

29 Mar 2023

The responses to all editor and reviewers comments have been addressed in the Rebuttal letter attached. Comments Response Line/page 

Editor 

1) Please ensure that your manuscript meets PLOS ONE's style requirements, including those for file naming

 Thank you for the observation, this has been revised throughout the manuscript. 

2) We noticed you have some minor occurrence of overlapping text with the following previous publication(s), which needs to be addressed

 These have been addressed on Page 2, line numbers 39 to 43 and Page 13, line numbers 205 to 207 

3) We suggest you thoroughly copyedit your manuscript for language usage, spelling, and grammar. If you do not know anyone who can help you do this, you may wish to consider employing a professional scientific editing service 

 This has been considered and done as suggested throughout the manuscript. 

 Thank you. All the requested files have been provided and uploaded as required. 

 The name of the copyeditor has also been added in the acknowledgement section on Page 13, line number 215

Upon re-submitting your revised manuscript, please upload your study’s minimal underlying data set as either Supporting Information files or to a stable, public repository and include the relevant URLs, DOIs, or accession numbers within your revised cover letter This has been submitted and also acknowledged by the reviewers in the initial submission, however, attached is the link to the public repository access to additional information. URL http://hdl.handle.net/10570/7471

http://makir.mak.ac.ug: Criterion based audit on the management of preterm birth in Mulago National Referral hospital

Reviewer 1 

1) The numbers mentioned as numerator and denominator in Table no. 2 are not clear. This should be reworded for clarity. 

Thank you for the comment. This has been revised and made clearer on Table 2, line numbers 141-142

 2) It is not clear what is meant by the following sentence on page no. 8," Since there was no documented protocol on management of pre-term birth at Mulago Hospital labor ward, the baseline audit was conducted as per the components derived by the team of experts". This needs to reworded for clarity 

 Rewording has been done for Clarity noted on Page 8, line numbers 133-138

 3) There are several grammatical errors that need to be corrected 

 Thank you for the keen observation. Grammatical errors have been corrected throughout the manuscript. 

Reviewer 2 

1) The rationale in abstract and introduction should be similar 

 Thank you for the keen observation. This has been revised to make sure that the rationales in the abstract and introduction are similar. Page 2, line numbers 24-25 

 2) Certain grammatical mistakes are there. Page 4 4th paragraph instead of using word "basing" write "on the basis of" or use another word. Ideally referring to table write the number in numerical form, Table 2 rather than table two. 

Thank you for the suggestion, the rewording has been done as per your suggestion. Table numbers in the text have also been revised to ensure that they are in numerical form throughout the manuscript. Page 4, line number 75 , page 6 line 129 , Page 8, line 139 , Page 10, line 152. 

3) There is early neonatal death in 1st audit that is given 5.8% the second audit done after observing protocol, the early neonatal death increased to 11% which is quite significant. What were the reasons please mention in discussion. 

4) The reasons for this have been added in the discussion. Page 11, line numbers 180- 184

Mention who were the staff that received the information of both audit results? were they the same as in the first audit or a different group These have been highlighted and mentioned on Page 9, line numbers 113-120, and Page 9 first paragraph rephrase the paragraph, long sentence used as well as structure of sentence is unclear. This paragraph has been rephrased as suggested Page 8, line numbers 133-138 

 5) Conclusion should reflect your study findings and not a general opinion or suggestion. 

This has been revised in the abstract and main text conclusions. Page 2, line numbers 35 & 36 and Page 13, line number 208

---

## [Editor Report · Decision Letter 1]

11 Apr 2023

Management of preterm birth using protocols in a low resource setting.

PONE-D-23-00862R1

Dear Dr. Akello

We’re pleased to inform you that your manuscript has been judged scientifically suitable for publication and will be formally accepted for publication once it meets all outstanding technical requirements.

Kind regards,

Lumaan Sheikh, FRCOG

Academic Editor

PLOS ONE
---

## [Editor Report · Acceptance letter]

14 Apr 2023

PONE-D-23-00862R1 

Management of preterm birth using protocols in a low resource setting. 

Dear Dr. Akello:

I'm pleased to inform you that your manuscript has been deemed suitable for publication in PLOS ONE. Congratulations! Your manuscript is now with our production department. 

Kind regards, 

on behalf of

Dr. Lumaan Sheikh 

Academic Editor

PLOS ONE